# Discovering Hidden Associations among Environmental Disclosure Themes Using Data Mining Approaches

Ece Acar [1], Görkem Sarıyer [1,*], Vipul Jain [2] and Bharti Ramtiyal [3]

1 Department of Business Administration, Faculty of Business, Yasar University, Izmir 35100, Turkey; ece.acar@yasar.edu.tr
2 Operations and Supply Chain Management, Wellington School of Business and Government, Wellington 6012, New Zealand; vipul.jain@vuw.ac.nz
3 Department of Management Studies, Graphic Era (Deemed to be University), Dehradun 248002, India; bharti.mnit2022@gmail.com
* Correspondence: gorkem.ataman@yasar.edu.tr

**Abstract:** Environmental concerns play a crucial role in sustainability and public opinion on supply chains. This is why, how, and to what extent the firms experience environmental-related actions and inform their stakeholders, which is under discussion by most researchers. This paper aims to leverage data mining and its capabilities by applying association rule mining to the environmental disclosure context. With the aim of extracting hidden relationships between environmental disclosure themes for BIST 100 firms serving the Turkish supply chain, this research implements a novel association rule mining approach and uses the Apriori algorithm. With this purpose, the environmental information of BIST 100 firms was collected manually from sustainability reports; the raw data were processed; and the following seven themes identified the representing firms' disclosure items: environmental management, climate change, energy management, emissions management, water management, waste management, and biodiversity management. The results indicate various hidden relations between the sector and disclosures, allowing us to generate sector-based rules between environmental disclosure themes.

**Keywords:** sustainability; environmental disclosure; data mining; association rule mining; supply chain

## 1. Introduction

"Climate change, global warming, natural disasters, pollution" are not just environmental problems; these are also social and economic problems [1,2]. Because, it is clear that most of the damage given to the environment in today's industrial world is caused by companies. Therefore, such blame puts companies under tremendous pressure to take environmentally responsible and often proactive actions. Also, with the increasing sensitivity to environmental problems around the world, the awareness of companies to respond to this sensitivity has increased [3]. To deal with this issue, companies chose to use environmental information disclosure as the most appropriate communication tool with their stakeholders. Environmental issues require a holistic and multifaceted approach. Therefore, not only is a supply chains' environmental disclosure adequate, but also governments need to take new actions. For instance, the Carbon Disclosure Project (CDP) was founded to incentivize companies and cities to disclose environmental information [4]. Over 6300 companies and 500 cities worldwide cooperate with CDP [5]. Greenhouse gas emissions, which increase ground-level temperatures, are thought to be the major factor contributing toward climate change. Therefore, to cope with this issue, the Kyoto Protocol of the United Nations Framework Convention on Climate Change entered with force in February 2005, aiming to limit greenhouse gas emissions [6]. Moreover, Chinese Government new environmental protection law (NEPL) in 2015 and French Government New Economic Regulations (NER)

in 2001 required from publicly traded firms to disclose nearly 60 indicators relating to their corporate social responsibility (CSR) activities in annual reports.

Due to the increasing concerns of society and governments, firms became exposed to the pressure and thus felt obliged to explain how responsible they were in their activities toward the environment, and that resulted in an increase in environmental disclosure. Disclosure, whether it is mandatory or voluntary or financial or non-financial, has been on the agenda of most academics and firms in recent years. It is an essential way for companies to express themselves. For years, disclosure, as is very well known, consisted of presenting quantitative information about the financial position and performance of companies through financial reports. However, as indicated above, in recent years, the direction of disclosure in corporate reports has shifted to narrative information that is voluntary and non-financial by nature, namely environmental information disclosure.

In the last decade, a great deal of academic research and notable practitioner literature have focused on the possibilities offered by data mining in the area of corporate environmental performance [7], sustainability [8,9], corporate social responsibility [10], and non-financial disclosure [11]. Data mining emerges as one of the most important current paradigms of advanced smart business analytics and decision support tools. This significant role of data mining is recognized by leading professional accountancy bodies like the American Institute of Certified Public Accountants (AICPA) Institute of Internal Auditors (IIA). AICPA acknowledged data mining among the technologies of the future and IAA stated that the priority of research should be focused on data mining [12].

Recent advances in technology have drawn the attention of researchers and practitioners to data mining and analytics; these provide opportunities for decision-makers to have a superior understanding of their organizations and make timely and efficient decisions [13]. Data mining refers to an integrative process that collects, manages, and examines huge volumes and varieties of data. It can also be defined as the process of deep analysis of large data sets to create valuable knowledge [14]. This technology enables us to take data into action, transform them into valuable information, and thus support operational and strategic decision-making processes [15]. Nowadays, created knowledge with the use of these technologies is considered one of the most important assets of supply chain companies [16].

The four main tasks of data mining are as follows: clustering, prediction, classification, and association. Among those tasks, specifically, clustering and prediction are used the most in accounting literature [17–19]. However, association analysis, namely association rule mining (ARM), which aims to explore hidden relations among variables, has not been studied much in the literature. ARM aims to create a structured method by discovering all the frequent patterns in a data set, creating remarkable rules from them, and thus using it as a tool to discover the hidden relations between items in data sets.

Amani and Fadlalla [20] conducted a review of the literature focusing on data mining applications in accounting. Their review revealed that 67% of data mining applications performed in accounting focused on classification, 12% on estimation, 6% on clustering, 5% on optimization, and just 2.5% focused on association.

In this context, unlike previous literature, this study aims to discover the hidden relations among environmental disclosure themes using ARM. To the best of our knowledge, this is a pioneering study focusing on environmental disclosure by taking advantage of the ARM technique and its capabilities. To this end, this research expands the application areas of ARM by using these technologies to discover the hidden relationships in supply chain companies' disclosure data. With the conceptualized themes of environmental disclosure, this research may also guide complex management decisions regarding environmental issues. By including both manufacturing and non-manufacturing companies and investigating the sector-based hidden patterns generated for the two sectors comparatively, this study not only differs from existing studies but also provides useful results and implications for practice.

The remainder of the paper is organized as follows: Section 2 briefly explains the environmental disclosure literature and data mining studies in the accounting literature.

Section 3 defines the methodological background. Section 4 describes the research method. Section 5 shows the results and interprets them. Section 6 provides a detailed discussion of the findings and presents concluding remarks together with limitations and further research comments.

## 2. Literature Review

Environmental disclosure is influenced by various factors. The nature of business activities, environmental performance of the firms, firm size and organizational visibility, ownership structure, firm resources, and board composition are among the first to appear [21–23]. Disclosure of environmental, social, and governance information has become a key component of investors' investment decisions [24]. Not only are investors willing to be informed about these increased risks related to environmental issues, but also other stakeholders are demanding information regarding the environmental practices of firms [22]. Consequently, the tendency of companies to make voluntary disclosures has been increasing over the years.

Also, researchers' interest in the subject has increased dramatically. A body of literature has focused on environmental disclosure practices from different features. While some have examined the relationship between environmental disclosure and environmental performance [25–27] and financial performance [28,29], some have examined the relationship between environmental disclosure and corporate governance [30–33]. For instance, it has been argued that countries that have traditionally been strongly involved in environmental protection will have a greater corporate environmental concern, leading to more disclosure practices. [34]. Therefore, it is very clear that narrative voluntary reporting is shaped by country-specific factors. In the context of developing countries, particularly Turkish firms, Akbas and Canikli [35] found a significant increase in the level of environmental disclosure, which is mostly narrative information and changes across sectors. Kılıç and Kuzey [33] studied the extent of climate change disclosure in the Turkish banking industry and found an increase from 2010 to 2016. Again, Kılıç and Kuzey [36] examined corporate governance characteristic effects on carbon emission disclosure and found a positive relationship between them. Nevertheless, less attention has been given to environmental disclosure practices in Turkey, even though environmental disclosure is voluntary and country/region-specific factors affect voluntary disclosure practices.

In the accounting information system, financial and non-financial data are excessive, but information disclosure, especially about voluntary environmental information, is often lacking. This deficiency affects the effective role of accounting information in the financial decision support system. Using data and data technologies, financial decision support systems can effectively predict the future trend of corporate development. This may help create decision-making information for executive benefits, which will increase the competitiveness of businesses. While environmental disclosure serves as a broad field of research, studies mostly revolve around the above-mentioned issues. Although the importance of data mining is emphasized by accounting authorities and used in many areas of the business world, it is still not studied enough in the area.

One of the early papers was that of Coakley and Brown [37], which focused on the modeling concerns of neural networks in the area of accounting and finance and classified them according to the research question, type of output, and parametric nature of the model. Kloptchenko et al. [38] combined data and text mining techniques to analyze quantitative and qualitative contents of financial reports to reveal some indications about future financial performance. Similarly, Magnusson et al. [39] applied a multi-methodological approach and applied two different data mining techniques to the financial and accompanying textual data of the telecommunication sector to discover the developments reflected in the linguistic contents of the company's quarterly reports. Gaganis [40] used classification techniques to detect falsified financial statements using both financial and non-financial data. Hoffman and Lampe [19] used clustering to examine the balance sheet structure of supply chain firms. Tackett [41] explained the use of association rules to help auditors

find fraud patterns and relationships when they are not sure where to search. Alpar and Winkelstrater [42] discovered patterns of data quality issued in the form of association rules employed in data mining to accounting transactions. They aimed to discover potential data quality violations and find a balance between too many and too few transactions.

Moreover, a review study conducted by Amani and Fadlalla [20] revealed that 64% of data mining applications in accounting focused on assurance and compliance, 25% on managerial accounting, and 11% on financial accounting and accounting information systems. Among data mining applications, the most used task in assurance and compliance is classification (87%). In managerial accounting, the most common tasks used are estimation (42%), classification (25%), optimization (20%), forecasting (8%), association (4%), and exploration (2%). Finally, for financial accounting and accounting information systems, their review study revealed that 50% of data mining applications focus on classification and 35% on clustering. In line with all these data, it can be seen that classification and estimation related to data mining applications in accounting are used remarkably. They revealed that data mining applications in accounting are mainly focused on assurance and compliance, that their main purpose is estimation, and that their prominent function is classification.

However, this overwhelming trend and characterization show that accounting does not take full advantage of many other data mining tasks, such as patterns and association analysis. Therefore, this study aims to benefit from the association rule mining function of data mining techniques to discover hidden relations among environmental disclosure themes.

## 3. Methodological Background

### 3.1. Association Rule Mining (ARM)

Data mining is the process of transforming raw data stored in databases, data warehouses, or data repositories into useable knowledge. As one of the most prominent emerging technologies, data mining allows researchers and decision-makers to extract important knowledge from raw data and then turn them into action. Thus, data mining also refers to knowledge discovery from data.

ARM, a popular data mining method, is used for discovering hidden associations and interesting relationships based on frequent patterns in the data set. Market basket analysis is a typical ARM. By finding items often located together in customers' market baskets, decision-makers gain insights into their customers' buying behavior and efficiently use this knowledge in various decisions such as shelf arrangement, promotion planning, pricing, inventory management, and many others.

ARM allows for inductive theorizing, which can address contingency or moderated relationships, and does not rely on linearity and normality assumptions [43]. Since ARM is not sensitive to such assumptions and real-life data sets generally include complex relationships, such as nonlinear, quadratic, etc., ARM is receiving increasing attention in real-life applications.

ARM was presented by Agrawal et al. [44]. Let $T = \{T_1, T_2, \ldots, T_n\}$ set of n transactions and $I = \{I_1, I_2, \ldots, I_m\}$ set of m items. Each transaction is a subset of items, $T_i \subseteq I$. An association rule has a form $X \implies Y$ where $X \subset I$, $Y \subset I$, and $X \cap Y = \varnothing$. In this rule, $X$ is called the antecedent and $Y$ is the consequent. The set of items, $X$ and $Y$, is called the item set. ARM is a two-step process. In the first step, all frequent item sets are found based on the user-specified minimum support (min_sup) measure. If any item set occurs at least as frequently as this min_sup value in the data set, then this item set is extracted in this first step of ARM. In the second step, from these frequent item sets satisfying or exceeding the min_sup value, strong association rules hidden in the data set are generated. Although ARM is a two-step process, the overall performance of this method is evaluated based on the first step since the second step is less costly compared to the first one [14].

### 3.2. Apriori Algorithm

For extracting association rules to discover relationships and dependencies within the data set, the most widely known and implemented algorithm is Apriori. It has been imple-

mented in various contexts, such as manufacturing services [16,45], health services [46,47], financial services [48], and many others.

This algorithm was introduced by Agrawal and Srikant [49]. Apriori uses an iterative approach where l-item sets are used to explore (l + 1) item sets. First, the whole data set is searched to calculate the count of each item, and by recording the item's satisfying pre-specified min_support value, frequent 1-item sets are found. The resulting data set is represented by $L_1$. Next, this set is used to find a set of frequent 2-item sets, $L_2$. By using $L_2$, the set of frequent 3-item sets is extracted. These iterative steps continue until no more frequent l-item sets are found.

In the algorithmic process of Apriori, item set I of length m is frequent if and only if every subset of I with length m − 1 is also frequent. In this regard, the Apriori algorithm develops a significant reduction in the search space and allows rule discovery in computationally feasible time.

## 4. Methods

### 4.1. Data Set

This study covered the environmental disclosure data of BIST 100 (Borsa İstanbul, Istanbul, Türkiye) firms for the year 2020. Although 100 companies are listed in this index, since only non-financial companies were considered for our research purposes, 63 non-financial companies were used in the study. These companies were operating in various industries such as (1) paper and paper products and printing and publishing; (2) basic metal; (3) chemicals, petroleum rubber, and plastic products; (4) construction and public works; (5) electricity, gas, and steam; (6) fabricated metal products, machinery, electrical equipment, and vehicles; (7) food, beverage, and tobacco; (8) human health and social work activities; (9) mining and quarrying; (10) non-metallic mineral products; (11) technology; (12) textiles, wearing apparel, and leather; (13) transportation storage and telecommunication; and (14) wholesale and retail trade, restaurants, and hotels.

Following the examination of environmental information given in the reports of these companies, 7 themes were identified that represented the firms' disclosure items. Then, firms were coded "1" if they disclosed that item and "0" if otherwise. Table 1 summarizes these themes and their descriptions.

**Table 1.** Environmental disclosure themes.

| No. | Themes | Description |
|:---:|:---:|:---:|
| 1 | Environmental Management | Use of natural resources like soil, water, and air with environmentally acceptable practices. |
| 2 | Climate Change | Climate change, which is driven by greenhouse effects, refers to changes in climate together with global warming. |
| 3 | Energy Management | Correct management of natural resources to contribute to the lives of living things. |
| 4 | Emissions Management | Emissions management; reducing emissions, including greenhouse gases (GHG); and tackling climate change. |
| 5 | Water Management | The effort to transfer and sustain the resource, which is ultimately found in nature for the survival and development of living things, to future generations. |
| 6 | Waste Management | Recycling and energy production of foreign and waste materials released to nature after the consumption of living things and companies. |
| 7 | Biodiversity Management | Deliberate regulation of resources by humans to conserve biodiversity. |

### 4.2. General Concept

This paper proposes a novel data-driven approach to discover hidden associations between environmental disclosure themes. It combines the investigations from the spatial dimension (companies) and the descriptive dimension (analysis). As a five-stage process, the research framework is presented in Figure 1. In Stage 1, raw data including environmental disclosure activities and sector information of 63 non-financial companies listed in BIST 100 were collected. In Stage 2, the required preparation steps to obtain structured data set for further processing with ARM algorithms were accomplished. In this stage, companies that had missing or irrelevant data or the ones that did not disclose any information were excluded from the analysis. After these tasks were completed, a whole data set covering environmental disclosure activities and sector data of 50 companies was formed.

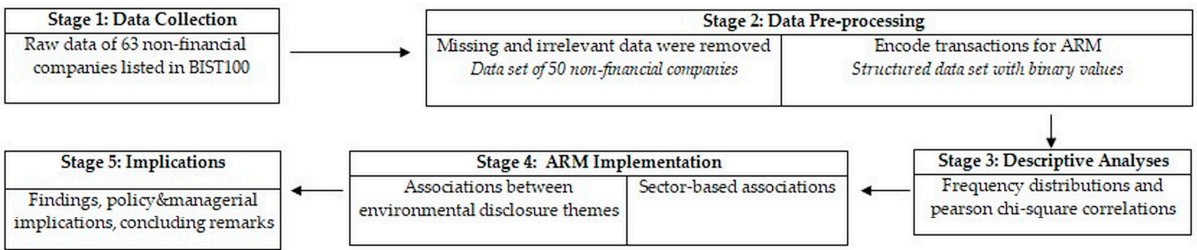

**Figure 1.** Research framework.

To convert this data set into a structured data set, which was ready for the ARM technique, transaction encoding was applied and the data set was transformed into a binary form. In Stage 3, by visualizing the data set and preparing frequency/percentage distributions, data were descriptively analyzed and relations between study variables were further explored based on Pearson chi-square analysis. Experimentation was handled with ARM implementations in Stage 4. In this stage, first association rules between only environmental disclosure themes were discovered, and then sector-based rules were extracted. Discovered rules were comparatively interpreted, and further implications for decision-makers were presented in light of these rules in Stage 5.

### 4.3. ARM Implementation

Data analysis and experimentation were performed in Python programming language. Required pre-processing steps were initially accomplished before the ARM implementation. With the use of the dropna () function in the pandas module (Version 1.1.5), the data set was cleaned from the missing values. Categorical conversion of the variables was implemented with the categorical class initializer of the pandas module (Version 1.1.5) to convert the variables that were categorical but represented in numbers. Transaction encoding was accomplished using the preprocessing package of the mlxtend module (Version 0.19.0) to convert the data set into a binary structure.

ARM was implemented using the Apriori algorithm. With the use of the frequent_patterns package of mlxtend module (Version 0.19.0) in Python and applying the Apriori algorithm, ARM was implemented to discover association rules in the data set. In Stage 3, by ignoring the role of the sector, sector-related data were hidden to extract associations only between environmental disclosure themes. In Stage 4, the whole data set, including sector information, was used to obtain sector-based associations between environmental disclosure themes.

### 4.4. Measures

A data-driven system implementing ARM is likely to produce countless patterns or rules, particularly for large data sets. However, only a limited portion of these patterns will be interesting and useful. The usefulness or interestingness of discovered pattern is measured based on some measures. The usefulness of the discovered rule depends on its comprehensibility, its accuracy to some extent, and its novelty [50]. Besides these subjective

measures, there exist three objective measures for evaluating discovered rules: support, confidence, and lift.

For an item set $X \subseteq I$, support $(X)$ is defined as the percentage of transactions $T_i \in T$, such that $X \subseteq T_i$. The rule $X \Longrightarrow Y$ holds in the transaction set $T$ with support s, where s is the percentage of transactions in $T$ including $X \cup Y$. Thus, $support(X \Longrightarrow Y) = P(X \cup Y)$. The discovered rule $X \Longrightarrow Y$ has a confidence c in transaction data set $T$, where c is the percentage of transactions in $T$ including $X$, which also contains $Y$. This is taken to be the conditional probability formalized as $confidence(X \Longrightarrow Y) = P(Y \backslash X)$ or $confidence(X \Longrightarrow Y) = support(X \cup Y)/support(X)$. Lift is the other measure in the evaluation of discovered rules. The occurrence of itemset $X$ is independent of the occurrence of itemset $Y$ if $P(X \cup Y) = P(X)P(Y)$; otherwise, item sets $X$ and $Y$ are dependent and correlated as events, and lift is calculated as $lift(X, Y) = P(X \cup Y)/(P(X)P(Y))$. If the resulting value of this lift measure is less than 1, then the occurrence of $X$ is negatively correlated with $Y$, and if it is greater than 1, $X$ and $Y$ are positively correlated. The calculated value of 1 for a lift measure implies that $X$ and $Y$ are independent and no relationship exists between them.

## 5. Results

### 5.1. Descriptive Analysis

The data set covered 50 companies' data. Frequency distributions based on sector and environmental themes are presented in Figure 2.

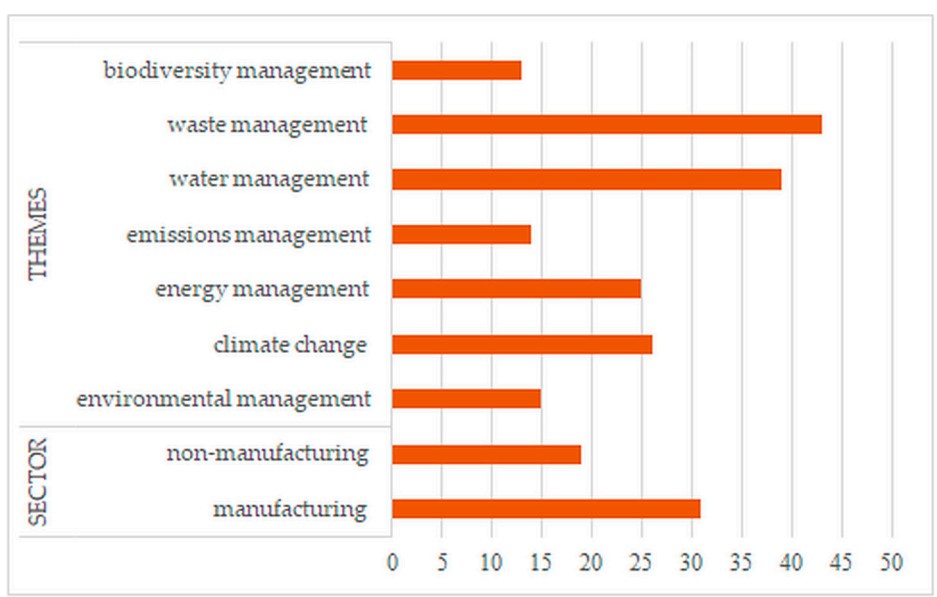

**Figure 2.** Number of companies disclosing information on each theme.

Figure 2 shows that while the majority of the study set's companies were manufacturing companies (62%), the remaining (38%) were non-manufacturing companies. While most of the companies made disclosures on waste management (86%) and water management (78%), only a few of the companies disclosed information on environmental management (30%), emissions management (28%), and biodiversity management (26%).

Table 2 shows the sector-based frequencies and percentages of the companies disclosing information on each theme. For simplicity, we use notations for theme 1 through theme 7 to denote seven different themes, and these notations are also introduced in Table 2.

In Table 2, although the percentage distributions of information disclosure on each theme have a similar pattern for both of the sectors, some noticeable differences can be observed. While a higher portion of the non-manufacturing companies disclosed information on climate change, energy management, water management, and waste management, these ratios were comparatively smaller for manufacturing companies. On the other hand, only

the environmental management and emissions management percentages of manufacturing companies disclosing information were slightly higher compared to the percentages of non-manufacturing companies.

**Table 2.** Sector-based frequencies of information disclosure on each theme.

| Themes | Abbreviation | Manufacturing | | Non-Manufacturing | |
|---|---|---|---|---|---|
| | | **31 Companies** | | **19 Companies** | |
| | | **N** | **%** | **n** | **%** |
| Environmental management | Env.Man. | 10 | 32.26 | 5 | 26.32 |
| Climate change | Cli.Chn. | 14 | 45.16 | 12 | 63.16 |
| Energy management | Eng.Man. | 14 | 45.16 | 11 | 57.89 |
| Emissions management | Emis.Man. | 9 | 29.03 | 5 | 26.32 |
| Water management | Wat.Man. | 23 | 74.19 | 16 | 84.21 |
| Waste management | Wast.Man. | 26 | 83.87 | 17 | 89.47 |
| Biodiversity management | Biod.Man. | 8 | 25.81 | 5 | 26.32 |

As a final analysis, the correlation between the study variables is presented. Since the study variables are nominal, i.e., sector (manufacturing/non-manufacturing) and each theme (yes/no), chi-square test statistics. The significance of the correlation (*p*-values) between each pair of variables is presented in Table 3.

**Table 3.** Significance of the relation (*p*-values) between study variables.

| | Env.Man. | Cli.Chn. | Eng.Man. | Emis.Man. | Wat.Man. | Wast.Man. | Biod.Man. |
|---|---|---|---|---|---|---|---|
| Sector | 0.454 | 0.173 | 0.28 | 0.551 | 0.322 | 0.457 | 0.61 |
| Env.Man. | | 0.004 | 0.108 | 0.12 | 0.184 | 0.248 | 0.163 |
| Cli.Chn. | | | 0.005 | 0.446 | 0.044 | 0.453 | 0.131 |
| Eng.Man. | | | | 0.173 | 0.5 | 0.5 | 0.26 |
| Emis.Man. | | | | | 0.636 | 0.643 | 0.264 |
| Wat.Man. | | | | | | 0.487 | 0.404 |
| Wast.Man. | | | | | | | 0.594 |

Table 3 mainly showed the absence of significant relations between the study variables for most of the study variable pairs. The sector did not seem to be significantly associated with any of the environmental disclosure themes. No significant relation was observed between the theme pairs except for (1) environmental management/climate change (*p*: 0.004); (2) climate change/energy management (*p*: 0.005); and (3) climate change/water management (*p*: 0.044). This result emphasized the importance of using data analytics and ARM to reveal hidden associations in the data set.

### 5.2. Association Rules between Environmental Themes

In this section, to extract the hidden association rules between environmental themes, sector information was not taken into consideration. Thus, based on the generated rules, we aimed to observe on which themes companies disclose information together. In the experimentation, minimum support was taken as 10% and minimum confidence as 60%. We also fixed the number of variables as a consequence of the rule at 1. Removing repeated and incomprehensible rules, we presented the top 20 rules with the highest support and confidence levels in Table 4.

**Table 4.** Association rules between environmental themes.

| Rule # | Association Rules | | Evaluation Measure | | |
|---|---|---|---|---|---|
| | Antecedent | Consequent | Support | Confidence | Lift |
| 1 | Wat.Man. = yes | Wast.Man. = yes | 0.68 | 0.87 | 1.01 |
| 2 | Cli.Chn. = yes | Wat.Man. = yes | 0.46 | 0.88 | 1.13 |
| 3 | Cli.Chn. = yes | Wast.Man. = yes | 0.46 | 0.88 | 1.03 |
| 4 | Cli.Chn. = yes & Wast.Man. = yes | Wat.Man. = yes | 0.42 | 0.91 | 1.17 |
| 5 | Biod.Man. = yes | Wat.Man. = yes | 0.22 | 0.85 | 1.08 |
| 6 | Emis.Man. = yes | Wat.Man. = yes | 0.22 | 0.79 | 1.01 |
| 7 | Env.Man. = yes | Eng.Man. = yes | 0.20 | 0.67 | 1.33 |
| 8 | Wast.Man. = yes & Biod.Man. = yes | Wat.Man. = yes | 0.18 | 0.82 | 1.05 |
| 9 | Biod.Man. = yes | Cli.Chn. = yes | 0.18 | 0.69 | 1.33 |
| 10 | Emis.Man. = yes | Eng.Man. = yes | 0.18 | 0.64 | 1.29 |
| 11 | Emis.Man. = yes & Cli.Chn. = yes | Wast.Man. = yes | 0.16 | 1.00 | 1.16 |
| 12 | Biod.Man. = yes & Cli.Chn. = yes | Wat.Man. = yes | 0.16 | 0.89 | 1.14 |
| 13 | Biod.Man. = yes & Cli.Chn. = yes | Wast.Man. = yes | 0.16 | 0.89 | 1.03 |
| 14 | Eng.Man. = yes & Emis.Man. = yes | Wast.Man. = yes | 0.16 | 0.89 | 1.03 |
| 15 | Emis.Man. = yes & Cli.Chn. = yes & Wat.Man. = yes | Wast.Man. = yes | 0.14 | 1.00 | 1.16 |
| 16 | Eng.Man. = yes & Cli.Chn. = yes | Wat.Man. = yes | 0.14 | 0.88 | 1.12 |
| 17 | Wast.Man. = yes & Biod.Man. = yes & Cli.Chn. = yes | Wat.Man. = yes | 0.14 | 0.88 | 1.12 |
| 18 | Eng.Man. = yes & Cli.Chn. = yes | Wast.Man. = yes | 0.14 | 0.88 | 1.01 |
| 19 | Wast.Man. = yes & Eng.Man. = yes & Cli.Chn. = yes | Wat.Man. = yes | 0.12 | 0.86 | 1.09 |
| 20 | Wat.Man. = yes & Env.Man. = yes | Eng.Man. = yes | 0.12 | 0.6 | 1.2 |

Although reading and interpreting all of these top 20 rules is not required, just for clarity, we can discuss rules #1 and #11 as examples. Rule #1 says that most of the companies that disclosed information on water management also disclosed information on waste management, with respective support and confidence levels of 0.68 and 0.87. Thus rule #1 shows that there exists a hidden association between water and waste management themes. Similarly, rule #11 presents that if a company made information disclosures on climate change and emissions management themes, then this company most probably disclosed information on waste management themes, with respective confidence and support levels of 0.16 and 1. Thus, rule #11 highlights the hidden relationship between climate change, emission management, and waste management. In general, although a previous analysis (Table 3) showed that no significant relationship exists between most of the environmental disclosure theme pairs, according to the results of Table 4, we showed that hidden associations exist between these themes.

*5.3. Sector-Based Association Rules*

In this section, to discover sector-based associations between the environmental themes, we also used sector information of the study companies. The extracted rules allow us to make a comparative analysis between manufacturing and non-manufacturing companies. To extract 25 rules, the minimum support level decreased to 8% here, while keeping the other parameters the same as in the previous section. The generated sector-based rules are presented in Table 5.

**Table 5.** Sector-based association rules between environmental themes.

| Rule # | Association Rules | | Evaluation Measure | | |
|---|---|---|---|---|---|
| | **Antecedent** | **Consequent** | **Support** | **Confidence** | **Lift** |
| 1 | sector = manufacturing & Wat.Man. = yes | Wast.Man. = yes | 0.4 | 0.87 | 1.01 |
| 2 | sector = non-manufacturing & Wat.Man. = yes | Wast.Man. = yes | 0.28 | 0.88 | 1.02 |
| 3 | sector = manufacturing & Cli.Chn. = yes | Wat.Man. = yes | 0.26 | 0.93 | 1.19 |
| 4 | sector = non-manufacturing & Cli.Chn. = yes | Wast.Man. = yes | 0.24 | 1.00 | 1.16 |
| 5 | sector = manufacturing & Wast.Man. = yes & Cli.Chn. = yes | Wat.Man. = yes | 0.22 | 1.00 | 1.28 |
| 6 | sector = non-manufacturing & Cli.Chn. = yes | Wat.Man. = yes | 0.2 | 0.83 | 1.07 |
| 7 | sector = non-manufacturing & Wast.Man. = yes & Cli.Chn. = yes | Wat.Man. = yes | 0.2 | 0.83 | 1.07 |
| 8 | sector = manufacturing & Env.Man. = yes | Wast.Man. = yes | 0.18 | 0.90 | 1.05 |
| 9 | sector = manufacturing & Eng.Man. = yes & Wat.Man. = yes | Wast.Man. = yes | 0.18 | 0.90 | 1.05 |
| 10 | sector = non-manufacturing & Eng.Man. = yes | Wat.Man. = yes | 0.18 | 0.82 | 1.05 |
| 11 | sector = non-manufacturing & Wast.Man. = yes | Eng.Man. = yes | 0.18 | 0.53 | 1.06 |
| 12 | sector = manufacturing & Emis.Man. = yes | Wast.Man. = yes | 0.16 | 0.89 | 1.03 |
| 13 | sector = manufacturing & Wat.Man. = yes & Env.Man. = yes | Wast.Man. = yes | 0.14 | 1.00 | 1.16 |
| 14 | sector = manufacturing & Biod.Man. = yes | Wat.Man. = yes | 0.14 | 0.88 | 1.12 |
| 15 | sector = manufacturing & Biod.Man. = yes | Wast.Man. = yes | 0.14 | 0.88 | 1.02 |
| 16 | sector = non-manufacturing & Wast.Man. = yes & Wat.Man. = yes | Eng.Man. = yes | 0.14 | 0.50 | 1.00 |
| 17 | sector = manufacturing & Eng.Man. = yes & Emis.Man. = yes | Wast.Man. = yes | 0.12 | 1.00 | 1.16 |
| 18 | sector = manufacturing & Biod.Man. = yes & Wat.Man. = yes | Cli.Chn. = yes | 0.12 | 0.86 | 1.65 |
| 19 | sector = manufacturing & Wast.Man. = yes & Biod.Man. = yes | Wat.Man. = yes | 0.12 | 0.86 | 1.10 |
| 20 | sector = manufacturing & Wast.Man. = yes & Emis.Man. = yes | Eng.Man. = yes | 0.12 | 0.75 | 1.50 |
| 21 | sector = non-manufacturing & Env.Man. = yes | Eng.Man. = yes | 0.10 | 1.00 | 2.00 |
| 22 | sector = manufacturing & Wast.Man. = yes & Biod.Man. = yes & Cli.Chn. = yes | Wat.Man. = yes | 0.10 | 1.00 | 1.28 |
| 23 | sector = non-manufacturing & Wast.Man. = yes & Emis.Man. = yes | Cli.Chn. = yes | 0.08 | 1.00 | 1.92 |
| 24 | sector = manufacturing & Wast.Man. = yes & Emis.Man. = yes & Cli.Chn. = yes | Wat.Man. = yes | 0.08 | 1.00 | 1.28 |
| 25 | sector = manufacturing & Eng.Man. = yes & Cli.Chn. = yes | Wat.Man. = yes | 0.08 | 1.00 | 1.28 |

Similarly, we can interpret some instance rules from Table 5. For example, rules #1 and #2, respectively, showed that for both manufacturing and non-manufacturing-type companies, if a company disclosed a water management theme, then it is very likely to disclose information on waste management. In another instance, rule #8 showed that most of the manufacturing companies disclosing information on environmental management themes also disclosed on waste management themes. Rule #15 shows that most of the manufacturing

companies that disclose information on biodiversity management are very likely to disclose information on waste management. Rule #23 shows that for non-manufacturing companies, there exists a hidden association between climate change, emission management, and waste management themes. We, therefore, observed that, although sector and environmental information disclosure did not seem to be related based on chi-square test results, there exist various hidden relations between sector and information disclosure. In light of these hidden associations, we generated sector-based rules between environmental disclosure themes.

## 6. Discussion and Conclusions

In terms of the supply chain, if an annual report includes environmental disclosure, the target group of the information is the capital markets and investors, as the level of environmental disclosure is affected by stakeholders' demands. However, considering the firms' corporate reputation, environmental information must be adequate to serve suppliers and customers, which are the other groups in the supply chain. Regarding all sample firms are part of the supply chain, and on the other hand, they must Since the requirements of stakeholders in that chain, environmental information becomes crucial for the sustainability of companies. Therefore, in this study, we aimed to put forth firms' environmental disclosure levels and explore the hidden relations between environmental disclosure themes.

Using a sample of 50 companies from the BIST 100 sustainability-indexed companies for the year 2020, various hidden relations between the sector and environmental disclosure were documented, which means that we can generate sector-based rules between environmental disclosure themes. First of all, overall, our findings indicate that 86% of the firms disclosed waste management and 78% of the firms disclosed water management. This may be because most of the sample firms (62%) operate in the manufacturing industry. However, environmental management, emissions management, and biodiversity management disclosures do not seem to be given enough importance. Firms that disclosed those themes were 30%, 28%, and 26%, respectively. However, considering that disclosure is a holistic structure and each theme is more important than the other, we can conclude that firms need to show a similar sensitivity to environmental, emission, and biodiversity management. When we compare manufacturing and non-manufacturing firms' disclosure, non-manufacturing firms made relatively higher disclosure in five themes out of seven. Considering that the environmental impacts of manufacturing companies are higher than those of non-manufacturing companies, this result is quite surprising. However, we can explain this result because environmental performance affects environmental disclosure levels based on the nature of the industry in which the company is operating. For instance, firms in the mining industry disclose a considerable amount of water use and improve their understanding of how the mining industry interacts with water resources [51].

One of the key findings of our study is that, even though descriptive statistics show no relation between the sector and environmental information disclosure when we applied association rule mining, we uncovered various hidden relations between those above. For this, first of all, we ignored sector information and applied association rule mining to reveal the themes that companies disclose information together. The results showed that, for instance, water management and waste management, climate change and water management, climate change and waste management, etc., all had hidden relations. As a further step, we considered sector information and tried to discover sector-based association rules between the disclosure themes. Both for manufacturing and non-manufacturing firms, the results showed that there exist hidden relations between water and waste management. The fact that most of the firms make the disclosures in these two themes (86% and 78%) and that there is a hidden relationship between these two themes when the sector information is ignored proves the accuracy of this result. Also, for manufacturing companies, there exists a hidden relationship between climate change and water management. For non-manufacturing companies, we found the existence of a hidden relationship between climate change and waste management. Whether manufacturing or non-manufacturing, it is

inevitable that firms are already experiencing the effects of climate change, both in terms of increasing costs and interrupted production. That fact brings stakeholders in the supply chain to ask for more disclosure about the physical consequences of climate change, and this relationship is expected, given that water and waste management are also a part of climate change. Also, reducing information asymmetry with supply chain partners and driving sustainable development tends to encourage firms to disclose more information about climate change effects.

A significant caveat of our study is that, although the basic descriptive analysis seemed to show no significant relations between the disclosure themes, we applied association rule mining and found those hidden relations. This study allows firms to develop an understanding of sustainable environmental disclosure, showing that there are hidden relations between themes. Moreover, the results of our study allow managers to identify what the company needs to improve in its environmental disclosure policy regarding, for instance, the hidden relation between the usage of water sources and waste management. Decision-makers can guide their decisions about which themes are missing and interrelated and, therefore, what should be disclosed.

An inherent limitation of our study is the lack of firms disclosing environmental information due to its voluntary nature. Future research may consider this limitation to develop a better measurement and work with a sample that may be from other stock exchanges with higher numbers of company data. As another suggestion for future research, the generated rules can be validated with the updated data of the study's data set companies. We can also validate the generated rules using other AR algorithms such as FP-Growth and ECLAT. Additionally, in future research, we can implement this ARM-based data mining approach to discover hidden relations in other sustainability contexts.

**Author Contributions:** Conceptualization, E.A., V.J., and G.S.; literature review, E.A.; data collection, E.A.; methodology, G.S.; software, G.S. and B.R.; interpretation of results, G.S. and V.J.; discussion and conclusion, E.A, G.S. and B.R. All authors have read and agreed to the published version of the manuscript.

**Funding:** This research received no external funding.

**Data Availability Statement:** The data presented in this study are available on request from the corresponding author.

**Conflicts of Interest:** The authors declare no conflict of interest.

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
