# Peer review of "Discovering Hidden Associations among Environmental Disclosure Themes Using Data Mining Approaches"

_sustainability, doi:10.3390/su151411406_

Round 1

Reviewer 1 Report

In general, this study is relevant and evidence-based. However, the title of the manuscript indicates that the study covers the supply chain sector. And in the text of the manuscript itself, the sector of supply chains is practically not reflected in any way.

In addition, the title of the manuscript talks about big data.

However, it does not follow from the text of the manuscript that it was big data that was used. It remains not noted how much data was actually studied

Author Response

We would like to thank the esteemed referee for these comments. Although our research is not directly related to the supply chain, it includes the main companies that play an important role in the effective execution of inter-firm interaction and supply operations in Turkey. These companies have already been selected among the most traded companies in the Borsa Istanbul index.  By extracting the hidden relations between the environmental disclosure themes of these purposively selected companies, we believe this study may present implications ad open new research directions for supply chain sustainability research.

On the other hand, the data used in our study does not represent the definition of big data, as you stated. However, the method used to reveal hidden relationships in the between the environmental disclosure themes represents data-driven methods. As we have already presented in the findings section, although the basic statistical methods (Chi-square test of independence) show that there is almost no significant relationship between the determined themes, many hidden association rules have been revealed with this method.

Considering addressed two important comments , we have updated the title of our article as "Discovering Hidden Associations among Environmental Disclosure Themes Using Data Mining Approaches". In addition, we made the necessary explanations and revisions in the parts related to the supply chain and big data.

Reviewer 2 Report

The paper contains very interesting and topical issues, both in terms of science and business. In my opinion, the paper lacks correlation between the research methodology presented and the structure of the paper. Presentation of results at a very good level, I would suggest to dissect in more detail the directions for further research.

The text is free of errors in grammar and syntax. A good variety of professional vocabulary is used here. Language flows logically and is clear, understandable, and appropriate for the reader. The level of detail is consistent, length of sections is balanced, and headings and otherorganizational elements are consistent and marked appropriately.

Author Response

Thank you very much for your valuable comments and criticisms. We tried to improve our article by taking into account the comments of you and other reviewers. In our article, we aimed to determine the hidden relationships between sustainability themes by using the association rule mining function of data analytics. However, the definition and terminology, such as big data and big data-based approaches mentioned in some parts of the previous version of the article, created confusion about the methodology and led to inconsistencies throughout the article. We express our regret for this shortcoming caused by the flow of information among the authors working in writing different sections of the article and thank you for drawing attention to this point by examining our article with devotion. In this round, the methodology part, the title, the abstract, and the entire content, has been reviewed and revised in this context. In addition, the discussions on future studies in the conclusion part are also expanded and detailed.

Reviewer 3 Report

Dear Authors!

Thank you for giving me the opportunity to read and review the manuscript entitled „Discovering Supply Chain Sector Hidden Associations among Environmental Disclosure Themes Using Big Data-Driven Approaches". The manuscript is worth publishing and addresses an important issue company policies.

Title: The title represents the full manuscript.

Abstract: The abstract summarizes the whole study greatly, but it’s full of grammatical errors.   

Format: The format of the paper is acceptable, it follows the guideline, but some all of the figures must be redrawn (not good quality and unnecessary big).

Introduction and literature review: The introduction and goals are properly explained, the scientific gap can be recognized, but you should add some literature reviews to the introduction also, most of the sentences can be referred. The literature review is fully acceptable.

Materials and methods: The methods and algorithms used in a this paper are well known statistical and analytical tools. The novelty model easily understood and acceptable.  The measurement and limit numbers are well defined in my opinion.

Language: The language is mediocre, with a lot of typos and incorrect language, please have it checked by a linguist.  I find some typos and some of the elements in the text. See the attached file.

Conclusions: The conclusion is clear; the claims and proposals are proven professionally.

In my opinion, the article can be accepted after redrawing the figures and corrects the marked errors. I attached the file as a guide.

Language: The language is mediocre, with a lot of typos and incorrect language, please have it checked by a linguist.  I find some typos and some of the elements in the text. See the attached file.

Author Response

Thank you for taking the time to review our article and sharing your valuable feedback with us. In the light of your suggestions, we can summarize the changes we made in the article as follows:

-We changed the title to " Discovering Hidden Associations among Environmental Disclosure Themes Using Data Mining Approaches" based on the suggestions of the other referees.

-In order to eliminate the deficiencies that affect the spelling and English quality of the article, such as grammatical problems, misspellings, typos, and use of incorrect terminology, both in the abstract and in the entire text, all the authors have read the article by themselves, proofread it, and also passed it through the Grammarly-Pro program.

The literature gaps mentioned in the literature review section were also emphasized appropriately in the introduction.

-All the corrections mentioned in the supplementary file were applied in the revised form of the article (figures are redrawn, editing is applied, abbreviations are defined and used instead of theme1, thme2,..)

Reviewer 4 Report

The topic is interesting and has potential for publication.

The article is well structured and uses both classic and current references.

Improvement suggestion:

Summary: Clearly present the purpose of the study and the methodology used.

Introduction: There is a good contextualization of the problem. Present the research gap.

Methodology: Make clear the number corresponding to the population. It is also necessary to mention the size of the study sample.

Analysis of results/Discussion of results: Deepen comments on data analysis. It is interesting to compare their results with similar studies already published.

Make a grammatical revision in the text.

Author Response

Thank you for taking the time to review our article and sharing your valuable feedback with us. In the light of your suggestions, we can summarize the changes we made in the article as follows:

-to better present the study purpose and metodology in the abstract, we paraphrazed related sentence as “. This paper aims to leverage data minig and its capabilities by applying association rule mining to environmental disclosure context. With the aim of extracting hidden relationships between environmental disclosure themes for BIST 100 firms serving the Turkish supply chain, this research implements a novel association rule mining approach and uses the Apriori algorithm.”

-we presented the literature gap in the introduction section.

-in the methodology part, we detailed the presented information on the population, and sample.

-In order to eliminate the deficiencies that affect the spelling and English quality of the article, such as grammatical problems, misspellings, typos, and use of incorrect terminology, both in the abstract and in the entire text, all the authors have read the article by themselves, proofread it, and also passed it through the Grammarly-Pro program.

-we give detailed discussion on the obtained results.

-we revised and improved the figures

Round 2

Reviewer 1 Report

The changes made by the authors to the manuscript eliminated the comments. The manuscript is recommended for publication